# VN-Transformer: Rotation-Equivariant Attention for Vector Neurons

**Serge Assaad** *                                                serge.assaad@duke.edu
*Duke University*

**Carlton Downey**                                               cmdowney@waymo.com
*Waymo LLC*

**Rami Al-Rfou**                                                  rmyeid@waymo.com
*Waymo LLC*

**Nigamaa Nayakanti**                                            nigamaa@waymo.com
*Waymo LLC*

**Ben Sapp**                                                     bensapp@waymo.com
*Waymo LLC*

**Reviewed on OpenReview:** *https://openreview.net/forum?id=EiX2L4sDPG*

## Abstract

Rotation equivariance is a desirable property in many practical applications such as motion forecasting and 3D perception, where it can offer benefits like sample efficiency, better generalization, and robustness to input perturbations. Vector Neurons (VN) is a recently developed framework offering a simple yet effective approach for deriving rotation-equivariant analogs of standard machine learning operations by extending one-dimensional scalar neurons to three-dimensional "vector neurons." We introduce a novel "VN-Transformer" architecture to address several shortcomings of the current VN models. Our contributions are: $(i)$ we derive a rotation-equivariant attention mechanism which eliminates the need for the heavy feature preprocessing required by the original Vector Neurons models; $(ii)$ we extend the VN framework to support non-spatial attributes, expanding the applicability of these models to real-world datasets; $(iii)$ we derive a rotation-equivariant mechanism for multi-scale reduction of point-cloud resolution, greatly speeding up inference and training; $(iv)$ we show that small tradeoffs in equivariance ($\epsilon$-approximate equivariance) can be used to obtain large improvements in numerical stability and training robustness on accelerated hardware, and we bound the propagation of equivariance violations in our models. Finally, we apply our VN-Transformer to 3D shape classification and motion forecasting with compelling results.

---

*Work done during an internship at Waymo LLC.

# 1 Introduction

A chair – seen from the front, the back, the top, or the side – is still a chair. When driving a car, our driving behavior is independent of our direction of travel. These simple examples demonstrate how humans excel at using rotation invariance and equivariance to understand the world in context (see figure on the right). Unfortunately, typical machine learning models struggle to preserve equivariance/invariance when appropriate – it is indeed challenging to equip neural networks with the right inductive biases to represent 3D objects in an equivariant manner.

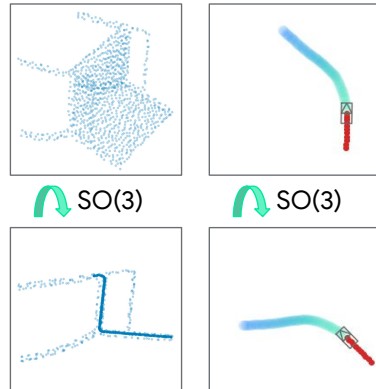

Modeling spatial data is a core component in many domains such as CAD, AR/VR, and medical imaging applications. In assistive robotics and autonomous vehicle applications, 3D object detection, tracking, and motion forecasting form the basis for how a robot interacts with humans in the real world. Preserving rotation invariance/equivariance can improve training time, reduce model size, and provide crucial guarantees about model performance in the presence of noise. Spatial data is often represented as a point-cloud data structure. These point-clouds require both permutation invariance and rotation equivariance to be modeled sufficiently well. Approaches addressing permutation invariance include Zaheer et al. (2018); Lee et al. (2019); Qi et al. (2017).

Recently, approaches jointly addressing rotation invariance and equivariance are gaining momentum. These can roughly be categorized into modeling invariance or equivariance by: *(i)* data augmentation, *(ii)* canonical pose estimation, and *(iii)* model construction. Approaches *(i)* and *(ii)* do not guarantee exact equivariance since they rely on model parameters to learn the right inductive biases (Qi et al., 2017; Esteves et al., 2018; Jaderberg et al., 2015; Chang et al., 2015). Further, data augmentation makes training more costly and errors in pose estimation propagate to downstream tasks, degrading model performance. Moreover, pose estimation requires labeling objects with their observed poses. In contrast, proposals in *(iii)* *do* provide equivariance guarantees, including the Tensor Field Networks of Thomas et al. (2018) and the SE(3)-Transformer of Fuchs et al. (2020) (see Section 2). However, their formulations require complex mathematical machinery or are limited to specific network architectures.

Most recently, Deng et al. (2021) proposed a simple and generalizable framework, dubbed *Vector Neurons* (VNs), that can be used to replace traditional building blocks of neural networks with rotation-equivariant analogs. The basis of the framework is lifting scalar neurons to 3-dimensional vectors, which admit simple mappings of SO(3) actions to latent spaces.

While Deng et al. (2021) developed a framework and basic layers, many issues required for practical deployment on real-world applications remain unaddressed. A summary of our contributions and their motivations are as follows:

**VN-Transformer.** The use of Transformers in deep learning has exploded in popularity in recent years as the de facto standard mechanism for learned soft attention over input and latent representations. They have enjoyed many successes in image and natural language understanding (Khan et al., 2022; Vaswani et al., 2017), and they have become an essential modeling component in most domains. Our primary contribution of this paper is to develop a VN formulation of soft attention by generalizing scalar inner-product based attention to matrix inner-products (*i.e.*, the Frobenius inner product). Thanks to Transformers' ability to model functions on sets (since they are permutation-equivariant), they are a natural fit to model functions on point-clouds. Our VN-Transformer possesses all the appealing properties that have made the original Transformer so successful, as well as rotation equivariance as an added benefit.

**Direct point-set input.** The original VN paper relied on edge convolution as a pre-processing step to capture local point-cloud structure. Such feature engineering is not data-driven and requires human involvement in designing and tuning. Moreover, the sparsity of these computations makes them slow to run

on accelerated hardware. Our proposed rotation-equivariant attention mechanism learns higher-level features directly from single points for arbitrary point-clouds (see Section 4).

**Handling points augmented with non-spatial attributes.** Real-world point-cloud datasets have complicated features sets – the $[x, y, z]$ spatial dimensions are typically augmented with crucial non-spatial attributes $[[x, y, z]; [a]]$ where $a$ can be high-dimensional. For example, Lidar point-clouds have intensity & elongation values associated with each point, multi-sensor point-clouds have modality types, and point-clouds with semantic type have semantic attributes. The VN framework restricted the scope of their work to spatial point-cloud data, limiting the applicability of their models for real-world point-clouds with attributes. We investigate two mechanisms to integrate attributes into equivariant models while preserving rotation equivariance (see Section 5).

**Equivariant multi-scale feature reduction.** Practical data structures such as Lidar point-clouds are extremely large, consisting of hundreds of objects each with potentially millions of points. To handle such computationally challenging situations, we design a rotation-equivariant mechanism for multi-scale reduction of point-cloud resolution. This mechanism learns how to pool the point set in a context-sensitive manner leading to a significant reduction in training and inference latency (see Section 6).

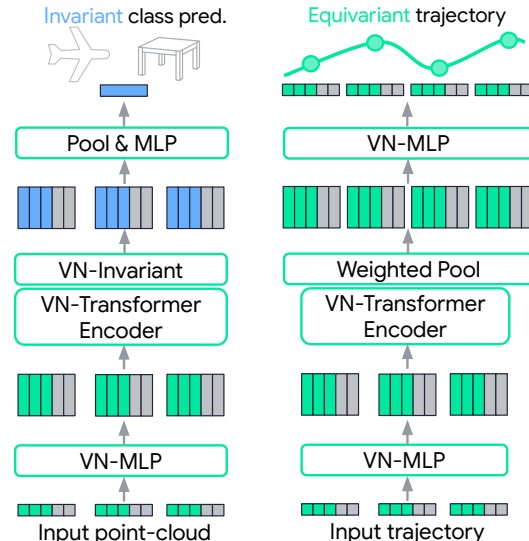

(a) Rotation-invariant classification model.

(b) Rotation-equivariant trajectory forecasting model.

Figure 1: VN-Transformer ("early fusion") models. Legend: (■) SO(3)-equivariant features; (■) SO(3)-invariant features; (■) Non-spatial features.

**$\epsilon$-approximate equivariance.** When attempting to scale up VN models for distributed accelerated hardware we observed significant numerical stability issues. We determined that these stemmed from a fundamental limitation of the original VN framework, where bias values could not be included in linear layers while preserving equivariance. We introduce the notion of $\epsilon$-approximate equivariance, and use it to show that small tradeoffs in equivariance can be controlled to obtain large improvements in numerical stability via the addition of small biases, improving robustness of training on accelerated hardware. Additionally, we theoretically bound the propagation of rotation equivariance violations in VN networks (see Section 7).

**Empirical analysis.** Finally, we evaluate our VN-Transformer on (*i*) the ModelNet40 shape classification task, (*ii*) a modified ModelNet40 which includes per-point non-spatial attributes, and (*iii*) a modified version of the Waymo Open Motion Dataset trajectory forecasting task (see Section 8).

## 2 Related work

The machine learning community has long been interested in building models that achieve equivariance to certain transformations, *e.g.*, permutations, translations, and rotations. For a thorough review, see Bronstein et al. (2021).

**Learned approximate transformation invariance.** A very common approach is to learn robustness to input transforms via data augmentation (Zhou & Tuzel, 2018; Qi et al., 2018; Krizhevsky et al., 2012; Lang et al., 2019; Yang et al., 2018) or by explicitly predicting transforms to canonicalize pose (Jaderberg et al., 2015; Hinton et al., 2018; Esteves et al., 2018; Chang et al., 2015).

**Rotation-equivariant CNNs.** Recently, there has been specific interest in designing rotation-equivariant image models for 2D perception tasks (Cohen & Welling, 2016; Worrall et al., 2017; Marcos et al., 2017; Chidester et al., 2018). Worrall & Brostow (2018) extended this work to 3D perception, and Veeling et al. (2018) demonstrated the promise of rotation-equivariant models for medical images.

**Equivariant point-cloud models.** Thomas et al. (2018) proposed Tensor Field Networks (TFNs), which use tensor representations of point-cloud data, Clebsch-Gordan coefficients, and spherical harmonic filters to build rotation-equivariant models. Fuchs et al. (2020) propose an "SE(3)-Transformer" by adding an attention mechanism for TFNs. One of the key ideas behind this body of work is to create highly restricted weight matrices that commute with rotation operations by construction (*i.e.*, $WR = RW$). In contrast, we propose a simpler alternative: a "VN-Transformer" which guarantees equivariance for arbitrary weight matrices, removing the need for the complex mathematical machinery of the SE(3)-Transformer. For a detailed comparison with Fuchs et al. (2020), see Appendix A.

**Controllable approximate equivariance.** Finzi et al. (2021) proposed equivariant priors on model weight matrices to achieve approximate equivariance, and Wang et al. (2022) proposed a relaxed steerable 2D convolution along with soft equivariance regularization. In this work, we introduce the related notion of "$\epsilon$-approximate equivariance," achieved by adding biases with small and controllable norms. We theoretically bound the equivariance violation introduced by this bias, and we also bound how such violations propagate through deep VN networks.

**Non-spatial attributes.** TFNs and the SE(3)-Transformer account for non-spatial attributes associated with each point (*e.g.*, color, intensity), which they refer to as "type-0" features. In this work, we investigate two mechanisms (early & late fusion) to incorporate non-spatial data into the VN framework.

**Attention-based architectures.** Since the introduction of Transformers by Vaswani et al. (2017), self-attention and cross-attention mechanisms have provided powerful and versatile components which are propelling the field of natural language processing forward (Devlin et al., 2019; Liu et al., 2019; Lan et al., 2020; Yang et al., 2020). Lately, so-called "Vision Transformers" (Dosovitskiy et al., 2021; Khan et al., 2022) have had a similar impact on the field of computer vision, providing a compelling alternative to convolutional networks.

## 3 Background

### 3.1 Notation & preliminaries

**Dataset.** Suppose we have a dataset $\mathcal{D} \triangleq \{X_p, Y_p\}_{p=1}^P$, where $p \in \{1, \ldots, P\}$ is an index into a point-cloud/label pair $\{X_p, Y_p\}$ – we omit the subscript $p$ whenever it is unambiguous to do so. $X \in \mathcal{X} \subset \mathbb{R}^{N \times 3}$ is a single 3D point-cloud with $N$ points. In a classification problem, $Y \in \mathcal{Y} \subset \{1, \ldots, \kappa\}$, where $\kappa$ is the number of classes. In a regression problem, we might have $Y \in \mathcal{Y} \subset \mathbb{R}^{N_{\text{out}} \times S_{\text{out}}}$ where $N_{\text{out}}$ is the number of output points, and $S_{\text{out}}$ is the dimension of each output point (with $N_{\text{out}} = S_{\text{out}} = 1$ corresponding to univariate regression).

**Index notation.** We use "numpy-like" indexing of tensors. Assuming we have a tensor $Z \in \mathbb{R}^{A \times B \times C}$, we present some examples of this indexing scheme: $Z^{(a)} \in \mathbb{R}^{B \times C}$, $Z^{(:,:,c)} \in \mathbb{R}^{A \times B}$, $Z^{(a_{\text{lo}}:a_{\text{hi}})} \in \mathbb{R}^{(a_{\text{hi}} - a_{\text{lo}} + 1) \times B \times C}$.

**Rotations & weights.** Suppose we have a tensor $V \in \mathbb{R}^{N \times C \times 3}$ and a rotation matrix $R \in \text{SO}(3)$, where $\text{SO}(3)$ is the three-dimensional rotation group. We denote the "rotation" of the tensor by $VR \in \mathbb{R}^{N \times C \times 3}$, defined as: $(VR)^{(n)} \triangleq V^{(n)}R, \quad \forall n \in \{1, \ldots, N\}$ – in other words, the rotated tensor $VR$ is simply the concatenation of the $N$ individually rotated matrices $V^{(n)}R \in \mathbb{R}^{C \times 3}$. Additionally, if we have a matrix of weights $W \in \mathbb{R}^{C' \times C}$, we define the product $WV \in \mathbb{R}^{N \times C' \times 3}$ by $(WV)^{(n)} \triangleq WV^{(n)}$.

**Invariance and equivariance.**

**Definition 1** (Rotation Invariance). *$f : \mathcal{X} \to \mathcal{Y}$ is rotation-invariant if $\forall R \in \text{SO}(3), \ X \in \mathcal{X}, \ f(XR) = f(X)$.*

**Definition 2** (Rotation Equivariance). *$f : \mathcal{X} \to \mathcal{Y}$ is rotation-equivariant if $\forall R \in \text{SO}(3), \ X \in \mathcal{X}, \ f(XR) = f(X)R$.*

For simplicity, we defined invariance/equivariance as above instead of the more general $f(X\rho_X(g)) = f(X)\rho_Y(g)$, which requires background on group theory and representation theory.

**Proofs.** We defer proofs to Appendix C.

### 3.2 The Vector Neuron (VN) framework

In the Vector Neuron framework (Deng et al., 2021), the authors represent a single point (*e.g.*, in a hidden layer of a neural network) as a matrix $V^{(n)} \in \mathbb{R}^{C \times 3}$ (see inset figure), where $V \in \mathbb{R}^{N \times C \times 3}$ can be thought of as a tensor representation of the entire point-cloud. This representation allows for the design of SO(3)-equivariant analogs of standard neural network operations.

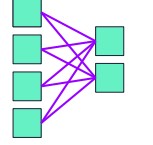 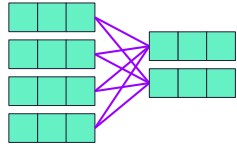

Classical neurons        Vector Neurons

**VN-Linear layer.** As an illustrative example, the VN-Linear layer is a function $\text{VN-Linear}(\,\cdot\,; W) : \mathbb{R}^{C \times 3} \to \mathbb{R}^{C' \times 3}$, defined by $\text{VN-Linear}(V^{(n)}; W) \triangleq W V^{(n)}$, where $W \in \mathbb{R}^{C' \times C}$ is a matrix of learnable weights. This operation is rotation-equivariant: $\text{VN-Linear}(V^{(n)}R; W) = W V^{(n)} R = (W V^{(n)}) R = \text{VN-Linear}(V^{(n)}; W) R$.

Deng et al. (2021) also develop VN analogs of common deep network layers ReLU, MLP, BatchNorm, and Pool. For further definitions and proofs of equivariance, see Appendix C. For further details, we point the reader to Deng et al. (2021).

## 4 The VN-Transformer

In this section, we extend the ideas presented in Deng et al. (2021) to design a "VN-Transformer" that enjoys the rotation equivariance property.

### 4.1 Rotation-invariant inner product

The notion of an inner product between tokens is central to the attention operation from the original Transformer (Vaswani et al., 2017). Consider the Frobenius inner product between two VN representations, defined below.

**Definition 3** (Frobenius inner product). *The Frobenius inner product between two matrices* $V^{(n)}, V^{(n')} \in \mathbb{R}^{C \times 3}$ *is defined by* $\langle V^{(n)}, V^{(n')} \rangle_F \triangleq \sum_{c=1}^{C} \sum_{s=1}^{3} V^{(n,c,s)} V^{(n',c,s)} = \sum_{c=1}^{C} V^{(n,c)} V^{(n',c)\intercal}$.

This choice of inner product is convenient because of its rotation invariance property, stated below.

**Proposition 1.** *The Frobenius inner product between Vector Neuron representations* $V^{(n)}, V^{(n')} \in \mathbb{R}^{C \times 3}$ *is rotation-invariant, i.e.* $\langle V^{(n)}R, V^{(n')}R \rangle_F = \langle V^{(n)}, V^{(n')} \rangle_F, \quad \forall R \in \text{SO}(3)$.

*Proof.*

$$\langle V^{(n)}R, V^{(n')}R \rangle_F = \sum_{c=1}^{C} (V^{(n,c)}R)(V^{(n',c)}R)^\intercal = \sum_{c=1}^{C} V^{(n,c)} RR^\intercal V^{(n',c)\intercal} = \sum_{c=1}^{C} V^{(n,c)} V^{(n',c)\intercal} = \langle V^{(n)}, V^{(n')} \rangle_F \quad (1)$$

$\square$

This rotation-invariant inner product between VN representations allows us to construct a rotation-equivariant attention operation, detailed in the next section.

### 4.2 Rotation-equivariant attention

Consider two tensors $Q \in \mathbb{R}^{M \times C \times 3}$ and $K \in \mathbb{R}^{N \times C \times 3}$, which can be thought of as sets of $M$ (resp. $N$) tokens, each a $C \times 3$ matrix. Using the Frobenius inner product, we can define an attention matrix $A(Q, K) \in \mathbb{R}^{M \times N}$ between the two sets as follows:

$$A(Q, K)^{(m)} \triangleq \text{softmax}\left( \frac{1}{\sqrt{3C}} \left[ \langle Q^{(m)}, K^{(n)} \rangle_F \right]_{n=1}^{N} \right), \tag{2}$$

This attention operation is rotation-equivariant w.r.t. simultaneous rotation of all inputs: Following Vaswani et al. (2017), we divide the inner products by $\sqrt{3C}$ since $Q^{(m)}, K^{(n)} \in \mathbb{R}^{C \times 3}$. From Proposition 1, $A(QR, KR) = A(Q, K) \ \ \forall R \in \mathrm{SO}(3)$.

Finally, we define the operation VN-Attn : $\mathbb{R}^{M \times C \times 3} \times \mathbb{R}^{N \times C \times 3} \times \mathbb{R}^{N \times C' \times 3} \to \mathbb{R}^{M \times C' \times 3}$ as:

$$\text{VN-Attn}(Q, K, Z)^{(m)} \triangleq \sum_{n=1}^{N} A(Q, K)^{(m,n)} Z^{(n)}. \tag{3}$$

**Proposition 2.** $\text{VN-Attn}(QR, KR, ZR) = \text{VN-Attn}(Q, K, Z)R.$

*Proof.*

$$\text{VN-Attn}(QR, KR, ZR)^{(m)} = \sum_{n=1}^{N} A(QR, KR)^{(m,n)} Z^{(n)} R \tag{4}$$

$$\overset{(*)}{=} \left[ \sum_{n=1}^{N} A(Q, K)^{(m,n)} Z^{(n)} \right] R = \text{VN-Attn}(Q, K, Z)^{(m)} R, \tag{5}$$

where $(*)$ holds since $A(QR, KR) = A(Q, K)$ (which follows straightforwardly from Proposition 1 and equation 2). $\square$

This is extendable to multi-head attention with $H$ heads, VN-MultiHeadAttn : $\mathbb{R}^{M \times C \times 3} \times \mathbb{R}^{N \times C \times 3} \times \mathbb{R}^{N \times C' \times 3} \to \mathbb{R}^{M \times C' \times 3}$:

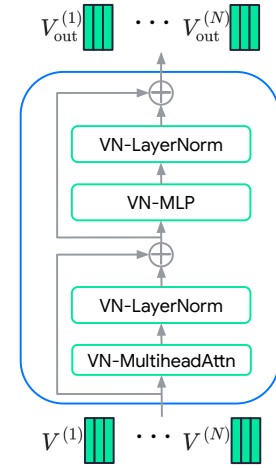

Figure 2: VN-Transformer encoder block architecture. VN-MultiheadAttn and VN-LayerNorm are defined in equation 6 and equation 7, respectively. VN-MLP is a composition of VN-Linear, VN-BatchNorm, and VN-ReLU layers from Deng et al. (2021).

$$\text{VN-MultiHeadAttn}(Q, K, Z) \triangleq W^O \left[ \text{VN-Attn}(W_h^Q Q, W_h^K K, W_h^Z Z) \right]_{h=1}^{H}, \tag{6}$$

where $W_h^Q, W_h^K \in \mathbb{R}^{P \times C}$, $W_h^Z \in \mathbb{R}^{P \times C'}$ are feature, key, and value projection matrices of the $h$-th head (respectively), and $W^O \in \mathbb{R}^{C' \times HP}$ is an output projection matrix (Vaswani et al., 2017).[1] VN-MultiHeadAttn is also rotation-equivariant, and is the key building block of our rotation-equivariant VN-Transformer.

We note that a similar idea was proposed in Fuchs et al. (2020) – namely, they use inner products between equivariant representations (obtained from the TFN framework of Thomas et al. (2018)) to create a rotation-invariant attention matrix and a rotation-equivariant attention mechanism. Our attention mechanism can be thought of as the same treatment applied to the VN framework. For a more detailed comparison between this work and the proposal of Fuchs et al. (2020), see Appendix A.

## 4.3 Rotation-equivariant layer normalization

Deng et al. (2021) allude to a rotation-equivariant version of the well-known layer normalization operation (Ba et al., 2016), but do not explicitly provide it – we do so here for completeness (see Figure 3):

$$\text{VN-LayerNorm}(V^{(n)}) \triangleq \tag{7}$$

$$\left[ \frac{V^{(n,c)}}{||V^{(n,c)}||_2} \right]_{c=1}^{C} \odot \text{LayerNorm}\left( \left[ ||V^{(n,c)}||_2 \right]_{c=1}^{C} \right) \mathbb{1}_{1 \times 3},$$

where $\odot$ is an elementwise product, LayerNorm: $\mathbb{R}^C \to \mathbb{R}^C$ is the layer normalization operation of Ba et al. (2016), and $\mathbb{1}_{1 \times 3}$ is a row-vector of ones.

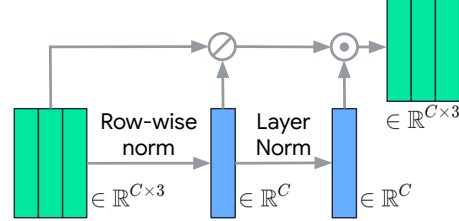

Figure 3: VN-LayerNorm: rotation-equivariant layer normalization. "Layer Norm" is the standard layer normalization operation of Ba et al. (2016). $\odot$ and $\oslash$ are row-wise multiplication and division.

---

[1] In practice, we set $H$ and $P$ such that $HP = C'$.

### 4.4 Encoder architecture

Figure 2 details the architecture of our proposed rotation-equivariant VN-Transformer encoder. The encoder is structurally identical to the original Transformer encoder of Vaswani et al. (2017), with each operation replaced by its rotation-equivariant VN analog.

## 5 Non-spatial attributes

"Real-world" point-clouds are typically augmented with crucial meta-data such as intensity & elongation for Lidar point-clouds and & sensor type for multi-sensor point-clouds. Handling such point-clouds while still satisfying equivariance/invariance w.r.t. spatial inputs would be useful for many applications.

We investigate two strategies (late fusion and early fusion) to handle non-spatial attributes while maintaining rotation equivariance/invariance w.r.t. spatial dimensions:

**Late fusion.** In this approach, we propose to incorporate non-spatial attributes into the model at a later stage, where we have already processed the spatial inputs in a rotation-equivariant fashion – our "late fusion" models for classification and trajectory prediction are shown in Figure 4.

**Early fusion.** Early fusion is a simple yet powerful way to process non-spatial attributes (Jaegle et al., 2021). In this approach, we do not treat non-spatial attributes differently (see Figure 1) – we simply concatenate spatial & non-spatial inputs before feeding them into the VN-Transformer. The VN representations obtained are $C \times (3 + d_A)$ matrices (instead of $C \times 3$).

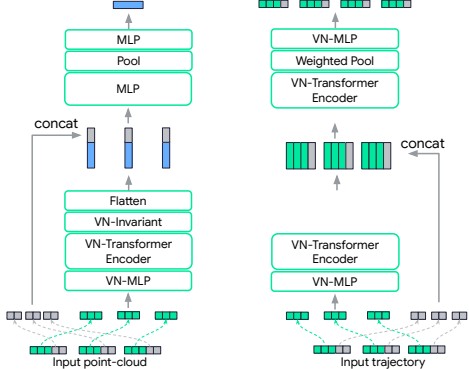

(a) Rotation-invariant classification model.
(b) Rotation-equivariant trajectory forecasting model.

Figure 4: VN-Transformer ("late fusion") models.
Legend: (■) SO(3)-equivariant features; (■) SO(3)-invariant features; (■) Non-spatial features.

## 6 Rotation-equivariant multi-scale feature aggregation

Jaegle et al. (2021) recently proposed an attention-based architecture, PerceiverIO, which reduces the computational complexity of the vanilla Transformer by reducing the number of tokens (and their dimension) in the intermediate representations of the network. They achieve this reduction by learning a set $Z \in \mathbb{R}^{M \times C'}$ of "latent features," which they use to perform QKV attention with the original input tokens $X \in \mathbb{R}^{N \times C}$ (with $M << N$ and $C' << C$). Finally, they perform self-attention operations on the resulting $M \times C'$ array, leading to a $\mathcal{O}(M^2 C')$ runtime instead of

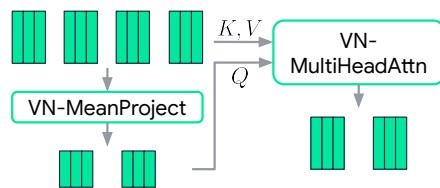

Figure 5: Rotation-equivariant latent features for Vector Neurons.

$\mathcal{O}(N^2 C)$ for each encoder self-attention operation, greatly improving time complexity during training and inference – a boon for time-critical applications such as real-time motion forecasting. However, such learnable latent features would violate equivariance in our case, since these learnable features would have no information about the original input's orientation. To remedy this, we instead propose to a learn a *transformation* from the inputs to the latent features (where the number of latent features is much smaller than the number of original inputs). Specifically, we propose to use a mean projection function VN-MeanProject$(V)^{(m)} \triangleq W^{(m)} \left[ \frac{1}{N} \sum_{n=1}^{N} V^{(n)} \right]$, where $W \in \mathbb{R}^{M \times C' \times C}$ is a learnable tensor. VN-MeanProject is both rotation-equivariant and permutation-invariant. We then perform VN-MultiHeadAttention between the resulting latent features and the original inputs $V$ to get a smaller set of VN representations. The architecture diagram for our proposed rotation-equivariant "latent feature" mechanism is shown in Figure 5.

## 7   $\epsilon$-approximate equivariance

We noticed that distributed training on accelerated hardware is numerically unstable for points with small norms. This is unique to VN models – the VN-Linear layer does not include a bias vector, which leads to frequent underflow issues on distributed accelerated hardware. We found that introducing small and controllable additive biases fixes these issues – we modify the VN-Linear layer by adding a bias with controllable norm:

$$\text{VN-LinearWithBias}(V^{(n)}; W, U, \epsilon) \triangleq WV^{(n)} + \epsilon U, \quad U^{(c)} \triangleq B^{(c)}/||B^{(c)}||_2, \tag{8}$$

where $\epsilon \geq 0$ is a hyperparameter controlling the bias norm, and $B \in \mathbb{R}^{C' \times 3}$ is a learnable matrix. This leads to significant improvements in training stability and model quality. In principle, VN-LinearWithBias is not equivariant, but its violation of equivariance can be bounded.

Work on equivariance by construction typically treats rotation equivariance as a binary idea – a model is either equivariant, or it is not. This can be relaxed by asking: how large is the violation of equivariance? We quantify this with the *equivariance violation* metric, defined by:

$$\Delta(f, X, R) \triangleq ||f(XR) - f(X)R||_F. \tag{9}$$

If $\Delta(f, X, R) \leq \epsilon$, we say $f$ is $\epsilon$-*approximately equivariant*.

We bound the equivariance violation of VN-LinearWithBias$(\cdot; W, U, \epsilon) : \mathbb{R}^{C \times 3} \to \mathbb{R}^{C' \times 3}$ below:

**Proposition 3.** VN-LinearWithBias *is* $(2\epsilon\sqrt{C'})$-*approximately equivariant (tight when $R = -I$).*

A natural next question is: how do such equivariance violations propagate through a deep model?

**Proposition 4.** *Suppose we have $K$ functions $f_k : \mathcal{X}_k \to \mathcal{X}_{k+1}$ (with $\mathcal{X}_k \subset \mathbb{R}^{C_k \times 3}, \mathcal{X}_{k+1} \subset \mathbb{R}^{C_{k+1} \times 3}$, $k \in \{1, \dots, K\}$) s.t.*

1. *$f_k$ is $\epsilon_k$-approximately equivariant for all $k \in \{1, \dots, K\}$*

2. *$f_k$ is $L_k$-Lipschitz (w.r.t. $|| \cdot ||_F$) for all $k \in \{2, \dots, K\}$.*

*Then, the composition $f_K \circ \cdots \circ f_1$ is $\epsilon_{1 \dots K}$-approximately equivariant, where*

$$\epsilon_{1 \dots K} \triangleq L_K(\cdots(L_3(L_2\epsilon_1 + \epsilon_2) + \epsilon_3) + \cdots) + \epsilon_K. \tag{10}$$

Intuitively, each layer $f_k$ "stretches" the equivariance violation error of the previous layers by its Lipschitz constant $L_k$, and adds its own violation $\epsilon_k$ to the total error.

## 8   Experiments

### 8.1   Rotation-invariant classification

Figure 1a shows our proposed VN-Transformer architecture for classification. It consists of rotation-equivariant operations (VN-MLP and VN-Transformer Encoder blocks), followed by an invariant operation (VN-Invariant), and finally standard Flatten/Pool/MLP operations to get class predictions. The resulting logits/class predictions are rotation-invariant.

Table 1: ModelNet40 test accuracy. Top block shows SO(3)-invariant baselines taken from Deng et al. (2021), included here for convenience.

| Model | Acc. | # Params |
|---|---|---|
| TFN (Thomas et al., 2018) | 88.5% | – |
| RI-Conv (Zhang et al., 2019) | 86.5% | – |
| GC-Conv (Zhang et al., 2020) | 89.0% | – |
| VN-PointNet (Deng et al., 2021) | 77.2% | 2.20M |
| VN-DGCNN (Deng et al., 2021) | 90.0% | 2.00M |
| VN-Transformer (ours) | 90.8% | 0.04M |

We evaluate our VN-Transformer classifier on the commonly used ModelNet40 dataset (Wu et al., 2015), a 40-class point-cloud classification problem. In Table 1, we compare our model with recent rotation-invariant models. The VN-Transformer outperforms the baseline VN models with orders of magnitude fewer parameters. Furthermore, we dispense with the computationally expensive edge-convolution used as a preprocessing step

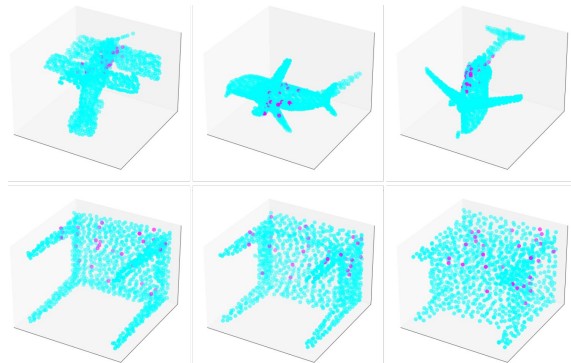 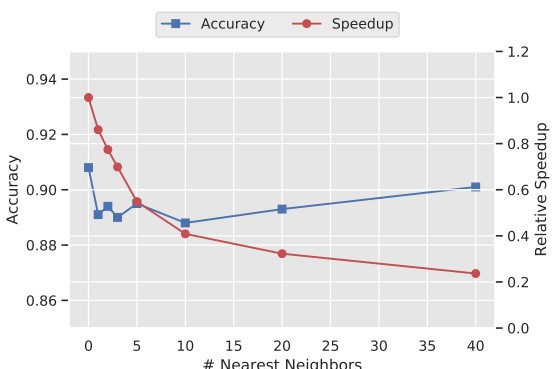

Figure 6: Example point-clouds from the ModelNet40 Polka-dot dataset. Cyan points correspond to $a_i = 0$, and pink points correspond to $a_i = 1$. Note that the "airplane" class has a narrower polka-dot radius than the "table" class, since we make the polka-dot radius dependent on the object class.

Figure 7: VN-Transformer ModelNet40 accuracy and relative training speed vs. number of nearest neighbors used in edge-convolution preprocessing. Speed is computed relative to zero neighbors (*i.e.*, no edge-convolution). Edge-convolution slows training speed by $\sim$5x and has little effect on model performance.

in the models of Deng et al. (2021) and find that the VN-Transformer's performance is relatively unaffected (see Figure 7).

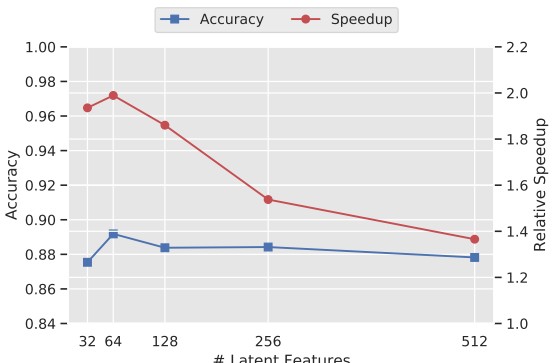

Figure 8: Test set accuracy on ModelNet40 vs. number of latent features (described in Figure 5). Latent features provide a $\sim$2x speedup with minimal acc. degradation (vs. row 3 of Table 2).

Table 2: Test set accuracy on ModelNet40 "Polka-Dot" dataset. Top block shows ModelNet40 results (*i.e.*, only spatial inputs). Bottom block shows ModelNet40 Polka-dot results. $\epsilon$ is the bias norm in VNLinearWithBias of eq. equation 8.

| Model | $\epsilon$ | Fusion | Features | Acc. |
|---|---|---|---|---|
| VN-PointNet | 0 | – | $[x, y, z]$ | 77.2% |
| VN-Transformer | 0 | – | $[x, y, z]$ | 88.5% |
| VN-Transformer | $10^{-6}$ | – | $[x, y, z]$ | 90.8% |
| VN-PointNet | 0 | Early | $[x, y, z, a]$ | 82.0% |
| VN-Transformer | 0 | Early | $[x, y, z, a]$ | 91.1% |
| VN-Transformer | $10^{-6}$ | Early | $[x, y, z, a]$ | **95.4%** |
| VN-Transformer | $10^{-6}$ | Late | $[x, y, z, a]$ | 91.0% |

## 8.2 Classification with non-spatial attributes

To evaluate our model's ability to handle non-spatial attributes, we design a modified version of the ModelNet40 dataset, called ModelNet40 "Polka-dot," which we construct (from ModelNet40) as follows: to each point $[x_i, y_i, z_i]$, we append $a_i \in \{0, 1\}$. Within a radius $r$ of a randomly chosen center, we randomly select 30 points and set $a_i = 1$. We make the "polka-dot" radius $r$ depend on the label $y \in \{1, \ldots, 40\}$ via $r(y) \triangleq r_{\text{lo}} + \frac{1}{39}(y - 1)(r_{\text{hi}} - r_{\text{lo}})$, where $r_{\text{lo}} = 0.3, r_{\text{hi}} = 1$. Figure 6 shows example point-clouds from the ModelNet40 Polka-dot dataset. By generating ModelNet40 Polka-dot in this way, we directly embed class information into the non-spatial attributes. In order to perform well on this task, models need to effectively fuse spatial and non-spatial information (there is no useful information in the non-spatial attributes alone since all point-clouds have $\sum_{i=1}^{N} a_i = 30$). The results on ModelNet40 Polka-dot are shown in Table 2. Both VN-PointNet and VN-Transformer benefit significantly from the binary polka-dots, suggesting they are able to effectively combine spatial and non-spatial information.

### 8.3 Latent features

Figure 8 shows our results on ModelNet40 when we reduce the number of tokens from 1024 (the number of points in the original point-cloud) to 32 using the latent feature mechanism presented in Figure 5. Using latent features provides a ~2x latency improvement (in training steps/sec) with minimal (~1.7%) accuracy degradation (compared with row 3 of Table 2). This suggests a real benefit of the latent feature mechanism in time-sensitive applications such as autonomous driving.

### 8.4 Rotation-equivariant motion forecasting

Figure 1b shows our proposed rotation-equivariant architecture for motion forecasting. In motion forecasting the goal is to predict the $[x, y, z]$ locations of an agent for a sequence of future timesteps, given as input the past locations of the agent. We evaluate the model on a simplified version of the Waymo Open Motion Dataset (WOMD; Ettinger et al., 2021):

- We select 4904 trajectories (3915 for training, 979 for testing).
- Each trajectory consists of 91 $[x, y, z]$ points for a single vehicle sampled at 5 Hz.
- We use the first 11 points (the past) as input and we predict the remaining 80 points (the future).

We evaluate the quality of our trajectory forecasting models using the Average Distance Error (ADE): $\text{ADE}(Y_i, \hat{Y}_i) \triangleq \frac{1}{T} \sum_{t=1}^{T} ||Y_i^{(t)} - \hat{Y}_i^{(t)}||_2$, where $T$ is the number of time-steps in the output trajectory and $Y_i, \hat{Y}_i \in \mathbb{R}^{T \times 3}$ are the ground-truth trajectory and the predicted trajectory, respectively. Results are shown in Table 3. Adding training-time random rotations about the $z$-axis yields improves the performance of the vanilla Transformer, and the VN-Transformer outperforms the vanilla Transformer (without the need for train-time rotation augmentations, thanks to equivariance). Figure 9 shows example predictions on WOMD. Equivariance violations of the vanilla Transformer models (columns (a) and (b)) are clearly demonstrated here, in contrast with the equivariant VN-Transformer (column (c)).

Table 3: Average Distance Error on WOMD. ↓ Lower is better. $\tilde{z}$ = random $z$-axis rotations used as data augmentation at training time. $\epsilon$ is the bias in the VNLinearWithBias layers (equation equation 8). The bottom block uses the speed $a$ as an added input feature (via early fusion).

| Model | $\epsilon$ | Features | ADE ($\downarrow$) |
|---|---|---|---|
| Transformer | – | $[x, y, z]$ | 5.01 |
| Transformer + $\tilde{z}$ | – | $[x, y, z]$ | 4.51 |
| VN-Transformer | 0 | $[x, y, z]$ | 4.91 |
| VN-Transformer | $10^{-6}$ | $[x, y, z]$ | **3.95** |
| VN-Transformer | 0 | $[x, y, z, a]$ | 5.01 |
| VN-Transformer | $10^{-6}$ | $[x, y, z, a]$ | **3.67** |

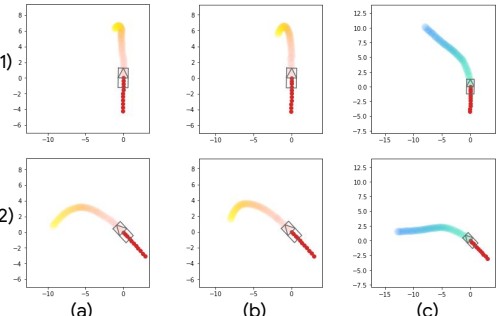

Figure 9: Example predictions on WOMD. Legend: Rectangle = current car position, Red points = input trajectory, Colored streaks = predicted trajectories. Columns are different trajectory models, with (a) Transformer, (b) Transformer + $\tilde{z}$ augmentations, and (c) VN-Transformer. Row (1) is one example in the dataset. Row (2) is a 45° rotation of the input points in Row (1).

## 9 Conclusion

In this paper, we introduced the VN-Transformer, a rotation-equivariant Transformer model based on the Vector Neurons framework. VN-Transformer is a significant step towards building powerful, modular, and easy-to-use models that have appealing equivariance properties for point-cloud data.

Limitations of our work include: ($i$) similar to previous work (Deng et al., 2021; Qi et al., 2017) we assume input data has been mean-centered. This is sensitive to outliers, and prevents us from making single-pass predictions for multi-object problems (we have to independently mean-center each agent first). Similarly, we have not addressed other types of invariance/equivariance (*e.g.*, scale invariance) in this work; ($ii$) Proposition 4 shows an error bound on the total equivariance violation of the network with $L_k$-Lipschitz layers. We know the Lipschitz constants of VN-Linear and VN-LinearWithBias (see Appendix C), but we have not yet determined them for other layers (*e.g.*, VN-ReLU, VN-MultiHeadAttn). We will address these gaps in future

work, and we will also leverage VN-Transformers to obtain state-of-the-art performance on a number of key benchmarks such as the full Waymo Open Motion Dataset and ScanObjectNN.

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

|  | VN-Transformer (ours) | SE(3)-Transformer (with $\ell = 1$) (Fuchs et al., 2020) |
|---|---|---|
| **Weight shape** | $W^Q \in \mathbb{R}^{C' \times C}$ ($C, C' = $ # of channels) | $W_Q^{11} \in \mathbb{R}^{3 \times 3}$ ($\in \mathbb{R}^{(2\ell'+1) \times (2\ell+1)}$ in the general case) |
| **Action on input** | Left-multiply: $W^Q V^{(n)}$ $\left( V^{(n)} \in \mathbb{R}^{C \times 3} \right)$ | Right-multiply: $f_i^1 W_Q^{11}$ $\left( f_i^1 \in \mathbb{R}^{1 \times 3} \right)$ |
| **Requirement for equivariance** | None, $W^Q \in \mathbb{R}^{C' \times C}$ is arbitrary | $W_Q^{11} = \sum_{J=0}^2 \sum_{m=-J}^J \varphi_J^{11}(\lVert x_i \rVert) Y_{Jm}(x_i / \lVert x_i \rVert) Q_{Jm}^{11}$, ($\varphi = $ radial neural net., $Y_{Jm} = $ spherical harmonics, $Q_{Jm}^{11} = $ Clebsch-Gordan coefficients) |

Table 4: Attention query computation in VN-Transformer vs. SE(3)-Transformer. $W^Q \in \mathbb{R}^{C' \times C}$ is the VN-Transformer weight matrix used to compute the query ($C$ and $C'$ are the number of input and query channels, respectively).

# A    Detailed comparison with SE(3)-Transformer (Fuchs et al., 2020)

## A.1    Attention computation

There is a rich literature on equivariant models using steerable kernels, and the SE(3)-Transformer is the closest development in this field to our work. Here, we make a detailed comparison between our work and the SE(3)-Transformer (and related steerable kernel-based models). For simplicity, we will compare the VN-Transformer with only spatial features vs. the SE(3)-Transformer with only type-1 features (i.e., spatial features).

**The key difference is in *the way the weight matrices are defined and how they interact with the input*.**

Specifically, Fuchs et al. use $3 \times 3$ weight matrices $W_Q^{11}, W_K^{11}, W_V^{11} \in \mathbb{R}^{3 \times 3}$ that act on the *spatial/representation* dimension of the input points (*e.g.*, via $f_i^1 W_Q^{11}$ where $f_i^1 \in \mathbb{R}^{1 \times 3}$).[2] As a result, in order to guarantee equivariance they need to design these matrices $W_Q^{11}, W_K^{11}, W_V^{11}$ such that they each commute with a rotation operation (in general $(f_i^1 R) W_Q^{11} \neq (f_i^1 W_Q^{11}) R$ – this depends on the choice of $W_Q^{11}$), hence the need for the machinery of Clebsch-Gordan coefficients, spherical harmonics, and radial neural nets to construct the weights.

In contrast, in our proposed attention mechanism, the matrices $W^Q, W^K, W^Z \in \mathbb{R}^{C' \times C}$ act on the *channel* dimension of the input (*e.g.*, via $W^Q V^{(n)}$, where $V^{(n)} \in \mathbb{R}^{C \times 3}$) and not the spatial dimension. As a result, the operations $W^Q V^{(n)}, W^K V^{(n)}, W^Z V^{(n)}$ are equivariant *no matter the choice of $W^Q, W^K, W^Z$*, since $W(V^{(n)} R) = (W V^{(n)}) R$. This results in a *significantly simpler* construction of rotation-equivariant attention that is (*i*) accessible to a wider audience (*i.e.*, it does not require an understanding of group theory, representation theory, spherical harmonics, Clebsch-Gordan coefficients, etc.) and (*ii*) much easier to implement.

For a side-by-side comparison of both attention query computations, see Table 4 above.

## A.2    VN-Linear vs. SE(3)-Transformer "self-interaction"

There is a relationship between the VN-Linear operation of Deng et al. (2021), and the "linear self-interaction" layers of Fuchs et al. (2020). Comparing equation (12) of Fuchs et al. (2020), repeated here for convenience:

$$\mathbf{f}_{\text{out},i,c'}^\ell = \sum_c w_{c'c}^{\ell\ell} \mathbf{f}_{\text{in},i,c}^\ell, \tag{11}$$

---

[2] $W_Q^{\ell\ell'}, W_K^{\ell\ell'}, W_V^{\ell\ell'} \in \mathbb{R}^{(2\ell+1) \times (2\ell'+1)}$ in the general case, where $\ell, \ell' \in \{0, 1, 2\}$ are feature types

with the VN-Linear operation of Deng et al. (2021):

$$V_{\text{out}}^{(n)} = W V^{(n)}, \quad W \in \mathbb{R}^{C' \times C}, V^{(n)} \in \mathbb{R}^{C \times 3}, \tag{12}$$

we see that these operations are identical.

However, our proposed VN-MultiHeadAttention is different and significantly simpler than the attention mechanism of the SE(3)-Transformer (see Section A.1), as it relies only on ($i$) the rotation-invariant Frobenius inner product and ($ii$) straightforward multiplication by an arbitary weight matrix to compute the keys, queries, and values (equivalent to VN-Linear/linear self-interaction). In that sense, it is closer in spirit to the original Transformer of Vaswani et al. (replacing vector inner product with Frobenius inner product). Further, as explained previously, it does not require the special construction of weight matrices using Clebsch-Gordan coefficients, spherical harmonics, and radial neural networks as in the SE(3)-Transformer.

## B  Experimental details

### B.1  Datasets

**ModelNet40**  The ModelNet40 dataset (Wu et al., 2015) is publicly available at `https://modelnet.cs.princeton.edu`, with the following comment under "Copyright":

*"All CAD models are downloaded from the Internet and the original authors hold the copyright of the CAD models. The label of the data was obtained by us via Amazon Mechanical Turk service and it is provided freely. This dataset is provided for the convenience of academic research only."*

**Waymo Open Motion Dataset**  The Waymo Open Motion Dataset (Ettinger et al., 2021) is publicly available at `https://waymo.com/open/data/motion/` under a non-commercial use license agreement. Full license details can be found here: `https://waymo.com/open/terms/`.

### B.2  Hyperparameter tuning

Table 5 shows the hyperparameters we swept over for all our experiments on ModelNet40, ModelNet40 Polka-dot, and the Waymo Open Motion Dataset.

| Hyperparameter | Value/Range |
| --- | --- |
| Feature dimension of VN-Transformer | {32, 64, 128, 256, 512, 1024} |
| Number of attention heads | {4, 8, 16, 32, 64, 128} |
| Hidden layer dimension in encoder's VN-MLP | {32, 64, 128, 256, 512} |
| Learning rate | $10^{-3}$ |
| Learning rate schedule | Linear decay |
| Optimizer | AdamW (Loshchilov & Hutter, 2019) |
| Epochs | 4000 |
| $\epsilon$ of VN-LinearWithBias | {0, $10^{-6}$} |

Table 5: Model hyperparameter ranges for ModelNet40, ModelNet40 Polka-dot, and Waymo Open Motion Dataset.

### B.3  Compute infrastructure

We trained our models on TPU-v3 devices. which are accessible through Google Cloud. Our longest training jobs ran for less than 3 hours on 32 TPU cores.

## C   Proofs

In this section, for convenience we will treat all 3D vectors as row-vectors: *e.g.*, $x \in \mathbb{R}^{1 \times 3}$. We also note that, while all our proofs of invariance/equivariance use matrices $V^{(n)} \in \mathbb{R}^{C \times 3}$, they can all be trivially generalized to $V^{(n)} \in \mathbb{R}^{C \times S}$.

### C.1   Partial invariance & equivariance

Assuming the input point-cloud consists of spatial inputs $X \in \mathcal{X} \subset \mathbb{R}^{N \times 3}$ and associated non-spatial attributes $A \in \mathcal{A} \subset \mathbb{R}^{N \times d_A}$ ($d_A$ is the number of non-spatial attributes associated with each point), we would like our model $f : \mathcal{X} \times \mathcal{A} \to \mathcal{Y}$ to satisfy the following property:

**Definition 4** (Partial rotation invariance). *A model $f : \mathcal{X} \times \mathcal{A} \to \mathcal{Y}$ satisfies partial rotation invariance if $\forall\ X \in \mathcal{X},\ A \in \mathcal{A},\ R \in \mathrm{SO}(3),\quad f(XR, A) = f(X, A)$.*

**Definition 5** (Partial rotation equivariance). *A model $f : \mathcal{X} \times \mathcal{A} \to \mathcal{Y}$ (where $\mathcal{Y} \subset \mathbb{R}^{N_{out} \times (3 + d_A)}$) satisfies partial rotation equivariance if $\quad \forall\ X \in \mathcal{X},\ A \in \mathcal{A},\ R \in \mathrm{SO}(3),\quad f(XR, A)^{(:,:3)} = f(X, A)^{(:,:3)} R$.*

We show here that the models in Figure 1 satisfy partial rotation invariance and equivariance (respectively).

Consider two rotation matrices $R_{d_1 \times d_1} \in \mathrm{SO}(d_1)$ and $R_{d_2 \times d_2} \in \mathrm{SO}(d_2)$.

**Lemma 1.** *The matrix $R_{(d_1 + d_2) \times (d_1 + d_2)} \triangleq \begin{bmatrix} R_{d_1 \times d_1} & \mathbf{0}_{d_1 \times d_2} \\ \mathbf{0}_{d_2 \times d_1} & R_{d_2 \times d_2} \end{bmatrix}$ is a valid rotation matrix in $\mathrm{SO}(d_1 + d_2)$.*

*Proof.* We begin by showing that $R_{(d_1 + d_2) \times (d_1 + d_2)}^{\mathsf{T}} = R_{(d_1 + d_2) \times (d_1 + d_2)}^{-1}$. We first compute $R_{(d_1 + d_2) \times (d_1 + d_2)}^{\mathsf{T}}$:

$$R_{(d_1 + d_2) \times (d_1 + d_2)}^{\mathsf{T}} = \begin{bmatrix} R_{d_1 \times d_1} & \mathbf{0}_{d_1 \times d_2} \\ \mathbf{0}_{d_2 \times d_1} & R_{d_2 \times d_2} \end{bmatrix}^{\mathsf{T}} = \begin{bmatrix} R_{d_1 \times d_1}^{\mathsf{T}} & \mathbf{0}_{d_2 \times d_1}^{\mathsf{T}} \\ \mathbf{0}_{d_1 \times d_2}^{\mathsf{T}} & R_{d_2 \times d_2}^{\mathsf{T}} \end{bmatrix} \overset{(*)}{=} \begin{bmatrix} R_{d_1 \times d_1}^{-1} & \mathbf{0}_{d_1 \times d_2} \\ \mathbf{0}_{d_2 \times d_1} & R_{d_2 \times d_2}^{-1} \end{bmatrix}, \tag{13}$$

where $(*)$ holds since $R_{d_1 \times d_1} \in \mathrm{SO}(d_1)$ and $R_{d_2 \times d_2} \in \mathrm{SO}(d_2)$ by assumption.
Now, we compute $R_{(d_1 + d_2) \times (d_1 + d_2)}^{-1}$:

$$R_{(d_1 + d_2) \times (d_1 + d_2)}^{-1} = \begin{bmatrix} R_{d_1 \times d_1} & \mathbf{0}_{d_1 \times d_2} \\ \mathbf{0}_{d_2 \times d_1} & R_{d_2 \times d_2} \end{bmatrix}^{-1} \tag{14}$$

$$= \begin{bmatrix} \left[ R_{d_1 \times d_1} - 0_{d_1 \times d_2} R_{d_2 \times d_2}^{-1} 0_{d_2 \times d_1} \right]^{-1} & \mathbf{0}_{d_1 \times d_2} \\ \mathbf{0}_{d_2 \times d_1} & \left[ R_{d_2 \times d_2} - 0_{d_2 \times d_1} R_{d_1 \times d_1}^{-1} 0_{d_1 \times d_2} \right]^{-1} \end{bmatrix} \tag{15}$$

$$= \begin{bmatrix} R_{d_1 \times d_1}^{-1} & \mathbf{0}_{d_1 \times d_2} \\ \mathbf{0}_{d_2 \times d_1} & R_{d_2 \times d_2}^{-1} \end{bmatrix}. \tag{16}$$

Hence, we have that $R_{(d_1 + d_2) \times (d_1 + d_2)}^{-1} = R_{(d_1 + d_2) \times (d_1 + d_2)}^{\mathsf{T}}$. Finally, we show that $\det(R_{(d_1 + d_2) \times (d_1 + d_2)}) = 1$:

$$\det(R_{(d_1 + d_2) \times (d_1 + d_2)}) = \det\left( \begin{bmatrix} R_{d_1 \times d_1} & \mathbf{0}_{d_1 \times d_2} \\ \mathbf{0}_{d_2 \times d_1} & R_{d_2 \times d_2} \end{bmatrix} \right) = \det(R_{d_1 \times d_1}) \det(R_{d_2 \times d_2}) \overset{(*)}{=} 1 \cdot 1 = 1, \tag{17}$$

where $(*)$ holds since $R_{d_1 \times d_1} \in \mathrm{SO}(d_1)$ and $R_{d_2 \times d_2} \in \mathrm{SO}(d_2)$ by assumption. $\square$

**Proposition 5.** *The VN-Transformer model $f : \mathcal{X} \times \mathcal{A} \to \mathcal{Y}$ (where $\mathcal{Y} \subset \mathbb{R}^{\kappa}$) shown in Figure 1a satisfies partial rotation invariance.*

*Proof.*

- For convenience, we reparametrize the model $f$ as $f_{\text{concat}} : \mathbb{R}^{N \times (3 + d_A)} \to \mathbb{R}^{\kappa}$ (with $\kappa$ the number of object classes) where $f_{\text{concat}}([X, A]) = f(X, A)$. It then suffices to show that $f_{\text{concat}}([XR, A]) = f_{\text{concat}}([X, A])R$.

- First, note that $f_{\text{concat}}$ is $SO(3 + d_A)$-invariant, since it is composed of $SO(3 + d_A)$-equivariant operations followed by a $SO(3 + d_A)$-invariant operation.
- Consider the matrix $R_{(3+d_A)\times(3+d_A)} \triangleq \begin{bmatrix} R & \mathbf{0}_{3\times d_A} \\ \mathbf{0}_{d_A\times 3} & I_{d_A\times d_A} \end{bmatrix}$, where $R \in SO(3)$ is an arbitrary 3-dimensional rotation. From Lemma 1, $R_{(3+d_A)\times(3+d_A)} \in SO(3 + d_A)$.

$$f_{\text{concat}}([X, A]R_{(3+d_A)\times(3+d_A)}) \stackrel{(*)}{=} f_{\text{concat}}([X, A]) \tag{18}$$

$$\Rightarrow f_{\text{concat}}([XR + A\mathbf{0}_{d_A\times 3}, X\mathbf{0}_{3\times d_A} + AI_{d_A\times d_A}]) = f_{\text{concat}}([X, A]) \tag{19}$$

$$\Rightarrow f_{\text{concat}}([XR, A]) = f_{\text{concat}}([X, A]), \tag{20}$$

where $(*)$ holds from $SO(3 + d_A)$-invariance of $f_{\text{concat}}$.

$\square$

**Proposition 6.** *The VN-Transformer model $f : \mathcal{X} \times \mathcal{A} \to \mathcal{Y}$ (where $\mathcal{Y} \subset \mathbb{R}^{N_{out}\times(3+d_A)}$) shown in Figure 1b satisfies partial rotation equivariance.*

*Proof.*

- For convenience, we reparametrize the model $f$ as $f_{\text{concat}} : \mathbb{R}^{N\times(3+d_A)} \to \mathbb{R}^{N_{\text{out}}\times(3+d_A)}$ where $f_{\text{concat}}([X, A]) = f(X, A)$. It then suffices to show that $f_{\text{concat}}([XR, A])^{(:,:3)} = f_{\text{concat}}([X, A])^{(:,:3)}R$.
- First, note that $f_{\text{concat}}$ is $SO(3 + d_A)$-equivariant, since it is composed of $SO(3 + d_A)$-equivariant operations.
- Consider the matrix $R_{(3+d_A)\times(3+d_A)} \triangleq \begin{bmatrix} R & \mathbf{0}_{3\times d_A} \\ \mathbf{0}_{d_A\times 3} & I_{d_A\times d_A} \end{bmatrix}$, where $R \in SO(3)$ is an arbitrary 3-dimensional rotation. From Lemma 1, $R_{(3+d_A)\times(3+d_A)} \in SO(3 + d_A)$.

$$f_{\text{concat}}([X, A]R_{(3+d_A)\times(3+d_A)}) \stackrel{(*)}{=} f_{\text{concat}}([X, A])R_{(3+d_A)\times(3+d_A)} \tag{21}$$

$$\Rightarrow f_{\text{concat}}([XR + A\mathbf{0}_{d_A\times 3}, X\mathbf{0}_{3\times d_A} + AI_{d_A\times d_A}])$$
$$= \left[ f_{\text{concat}}([X, A])^{(:,:3)}R + f_{\text{concat}}([X, A])^{(:,4:)}\mathbf{0}_{d_A\times 3}, f_{\text{concat}}([X, A])^{(:,:3)}\mathbf{0}_{3\times d_A} + f_{\text{concat}}([X, A])^{(:,4:)}\mathbf{I}_{d_A\times d_A} \right] \tag{22}$$

$$\Rightarrow f_{\text{concat}}([XR, A]) = \left[ f_{\text{concat}}([X, A])^{(:,:3)}R, f_{\text{concat}}([X, A])^{(:,4:)} \right] \tag{23}$$

$$\Rightarrow f_{\text{concat}}([XR, A])^{(:,:3)} = f_{\text{concat}}([X, A])^{(:,:3)}R, \tag{24}$$

where $(*)$ holds from $SO(3 + d_A)$-equivariance of $f_{\text{concat}}$.

$\square$

## C.2 $\epsilon$-approximate equivariance

**Proposition 3** (Restated)**.** VN-LinearWithBias$(\cdot; W, U, \epsilon)$ *is $(2\epsilon\sqrt{C'})$-approximately equivariant. This bound is tight when $R = -I_{3\times 3}$.*

*Proof.* Set $f \triangleq \text{VN-LinearWithBias}(\cdot; W, U, \epsilon)$:

$$f(XR) - f(X)R = (WXR + \epsilon U) - (WX + \epsilon U)R \tag{25}$$

$$= \epsilon U - \epsilon UR \tag{26}$$

$$\Rightarrow \Delta(f, X, R)^2 = ||f(XR) - f(X)R||_F^2 \tag{27}$$

$$= \sum_{c=1}^{C'} ||\epsilon(U^{(c)} - U^{(c)}R)||_2^2 \tag{28}$$

$$= \epsilon^2 \sum_{c=1}^{C'} ||U^{(c)} - U^{(c)}R||_2^2 \tag{29}$$

$$= \epsilon^2 \sum_{c=1}^{C'} ||U^{(c)}||_2^2 + ||U^{(c)}R||_2^2 - 2U^{(c)}R^\mathsf{T}U^{(c)\mathsf{T}} \tag{30}$$

$$\leq \epsilon^2 \sum_{c=1}^{C'} ||U^{(c)}||_2^2 + ||U^{(c)}R||_2^2 + 2U^{(c)}U^{(c)\mathsf{T}} \tag{31}$$

$$= \epsilon^2 \sum_{c=1}^{C'} 4||U^{(c)}||_2^2 \tag{32}$$

$$= 4\epsilon^2 C' \tag{33}$$

$$\Rightarrow \Delta(f, X, R) \leq 2\epsilon\sqrt{C'}. \tag{34}$$

$\square$

**Lemma 2.** *Suppose*

1. *$f : \mathcal{X}_1 \to \mathcal{X}_2$ (with $\mathcal{X}_1 \subset \mathbb{R}^{C_1 \times 3}, \mathcal{X}_2 \subset \mathbb{R}^{C_2 \times 3}$) is $\epsilon_f$-approximately equivariant.*
2. *$g : \mathcal{X}_2 \to \mathcal{X}_3$ (with $\mathcal{X}_3 \subset \mathbb{R}^{C_3 \times 3}$) is $\epsilon_g$-approximately equivariant and $L_g$-Lipschitz (w.r.t. the Frobenius norm).*

*Then the composition $g \circ f : \mathcal{X}_1 \to \mathcal{X}_3$ is $(L_g\epsilon_f + \epsilon_g)$-approximately equivariant.*

*Proof.*

$$\Delta(g \circ f, X, R) = ||g(f(XR)) - g(f(X))R||_F \tag{35}$$

$$= ||g(f(XR)) - g(f(X)R) + g(f(X)R) - g(f(X))R||_F \tag{36}$$

$$\leq ||g(f(XR)) - g(f(X)R)||_F + ||g(f(X)R) - g(f(X))R||_F \tag{37}$$

$$\overset{(*)}{\leq} ||g(f(XR)) - g(f(X)R)||_F + \epsilon_g \tag{38}$$

$$\overset{(**)}{\leq} L_g||f(XR) - f(X)R||_F + \epsilon_g \tag{39}$$

$$= L_g\Delta(f, X, R) + \epsilon_g \tag{40}$$

$$\overset{(***)}{\leq} L_g\epsilon_f + \epsilon_g, \tag{41}$$

where $(*)$ holds from $\epsilon_g$-approximate equivariance of $g$, $(**)$ holds because $g$ is $L_g$-Lipschitz, and $(***)$ holds from $\epsilon_f$-approximate equivariance of $f$. $\square$

**Proposition 4** (Restated)**.** *Suppose we have $K$ functions $f_k : \mathcal{X}_k \to \mathcal{X}_{k+1}$ (with $\mathcal{X}_k \subset \mathbb{R}^{C_k \times 3}, \mathcal{X}_{k+1} \subset \mathbb{R}^{C_{k+1} \times 3}$) for $k \in \{1, \dots, K\}$, satisfying the following:*

1. *$f_k$ is $\epsilon_k$-approximately equivariant for all $k \in \{1, \dots, K\}$.*

2. *$f_k$ is $L_k$-Lipschitz (w.r.t. $||\cdot||_F$) for all $k \in \{2, \dots, K\}$.*

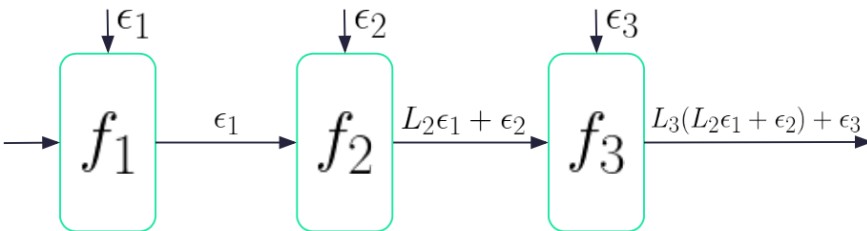

Figure 10: Violations of equivariance in a neural network with $L_k$-Lipschitz and $\epsilon_k$-approximately equivariant layers.

*Then, the composition $f_K \circ \cdots \circ f_1$ is $\epsilon_{1\ldots K}$-approximately equivariant, where:*

$$\epsilon_{1\ldots K} \triangleq L_K(\cdots(L_3(L_2\epsilon_1 + \epsilon_2) + \epsilon_3) + \cdots) + \epsilon_K \tag{42}$$

*Proof.*

$$\Delta(f_K \circ \cdots \circ f_1, X, R) = \Delta(f_K \circ (f_{K-1} \circ \cdots \circ f_1), X, R) \tag{43}$$

$$\overset{(K)}{\leq} L_K\Delta(f_{K-1} \circ \cdots \circ f_1, X, R) + \epsilon_K \tag{44}$$

$$\overset{(K-1)}{\leq} L_K(L_{K-1}(\Delta(f_{K-2} \circ \cdots \circ f_1, X, R) + \epsilon_{K-1}) + \epsilon_K \tag{45}$$

$$\vdots \tag{46}$$

$$\overset{(1)}{\leq} L_K(\cdots(L_3(L_2\Delta(f_1, X, R) + \epsilon_2) + \epsilon_3) + \cdots) + \epsilon_K \tag{47}$$

$$\leq L_K(\cdots(L_3(L_2\epsilon_1 + \epsilon_2) + \epsilon_3) + \cdots) + \epsilon_K \tag{48}$$

where $(K) - (1)$ hold from applying inequality equation 40 from Lemma 2 $K$ times (setting $g \triangleq f_k$ and $f \triangleq f_{k-1} \circ \ldots f_1$ at each step). $\qquad\square$

Figure 10 illustrates the propagation of equivariance violations through a composition of 3 functions.

**Proposition 7.** *The* VN-Linear$(\cdot; W) : \mathbb{R}^{C \times S} \to \mathbb{R}^{C' \times S}$ *layer is $\sigma(W)$-Lipschitz w.r.t. the Frobenius norm, where $\sigma(W)$ is the spectral norm of $W$. The same holds for* VN-LinearWithBias$(\cdot; W, U, \epsilon)$.

*Proof.* Consider $X_1, X_2 \in \mathbb{R}^{C \times S}$. We can write:

$$||WX_1 - WX_2||_F^2 = \sum_{s=1}^{S} ||WX_1^{(:,s)} - WX_2^{(:,s)}||_2^2 \tag{49}$$

$$= \sum_{s=1}^{S} ||W(X_1^{(:,s)} - X_2^{(:,s)})||_2^2 \tag{50}$$

$$\leq \sum_{s=1}^{S} \sigma^2(W)||X_1^{(:,s)} - X_2^{(:,s)}||_2^2 \tag{51}$$

$$= \sigma(W)^2 \sum_{s=1}^{S} ||X_1^{(:,s)} - X_2^{(:,s)}||_2^2 \tag{52}$$

$$= \sigma(W)^2||X_1 - X_2||_F^2 \tag{53}$$

$$\Rightarrow ||WX_1 - WX_2||_F \leq \sigma(W)||X_1 - X_2||_F \tag{54}$$

To see this for VN-LinearWithBias, note that $||(WX_1 + \epsilon U) - (WX_2 + \epsilon U)||_F^2 = ||WX_1 - WX_2||_F^2$ – we can then show the same result using the above proof. $\qquad\square$

## C.3 Equivariance of VN-LayerNorm

We define the VN analog of the layer normalization operation as follows:

$$\text{VN-LayerNorm}(V^{(n)}) \triangleq \left[\frac{V^{(n,c)}}{||V^{(n,c)}||_2}\right]_{c=1}^{C} \odot \text{LayerNorm}\left(\left[||V^{(n,c)}||_2\right]_{c=1}^{C}\right) \mathbb{1}_{1\times 3} \tag{55}$$

**Proposition 8.** $\text{VN-LayerNorm} : \mathbb{R}^{C\times 3} \to \mathbb{R}^{C\times 3}$ *is rotation-equivariant.*

*Proof.*

$$\text{VN-LayerNorm}(V^{(n)}R)^{(c)} = \frac{V^{(n,c)}R}{||V^{(n,c)}R||_2}\text{LayerNorm}\left(\left[||V^{(n,c)}R||_2\right]_{c'=1}^{C}\right)^{(c)} \tag{56}$$

$$\stackrel{(*)}{=} \frac{V^{(n,c)}R}{||V^{(n,c)}||_2}\text{LayerNorm}\left(\left[||V^{(n,c)}||_2\right]_{c'=1}^{C}\right)^{(c)} \tag{57}$$

$$= \left[\frac{V^{(n,c)}}{||V^{(n,c)}||_2}\text{LayerNorm}\left(\left[||V^{(n,c)}||_2\right]_{c'=1}^{C}\right)^{(c)}\right]R \tag{58}$$

$$= \text{VN-LayerNorm}(V^{(n)})^{(c)}R \tag{59}$$

$$= [\text{VN-LayerNorm}(V^{(n)})R]^{(c)} \tag{60}$$

$$\Rightarrow \text{VN-LayerNorm}(V^{(n)}R) = \text{VN-LayerNorm}(V^{(n)})R, \tag{61}$$

where $(*)$ holds from invariance of vector norms to rotations. $\square$

## C.4 Definitions of VN layers from Deng et al. (2021)

**VN-ReLU layer** The VN-ReLU layer is constructed as follows: from a given representation $V^{(n)} \in \mathbb{R}^{C\times 3}$, we compute a feature set $q \in \mathbb{R}^{C\times 3}$:

$$q \triangleq WV^{(n)}, \quad W \in \mathbb{R}^{C\times C}. \tag{62}$$

Then, we compute a set of $C$ "learnable directions" $k \in \mathbb{R}^{C\times 3}$:

$$k \triangleq UV^{(n)}, \quad U \in \mathbb{R}^{C\times C}. \tag{63}$$

Note that $W, U$ are learnable square matrices. Finally, we compute the output of the VN-ReLU operation $\text{VN-ReLU}(\cdot\,; W, U) : \mathbb{R}^{C\times 3} \to \mathbb{R}^{C\times 3}$ as follows:

$$\text{VN-ReLU}(V^{(n)})^{(c)} \triangleq \begin{cases} q^{(c)} & \text{if } \langle q^{(c)}, k^{(c)}\rangle \geq 0 \\ q^{(c)} - \langle q^{(c)}, \frac{k^{(c)}}{||k^{(c)}||}\rangle\frac{k^{(c)}}{||k^{(c)}||} & \text{o.w.} \end{cases} \tag{64}$$

Otherwise stated: if the inner product between the feature $q^{(c)}$ and the learnable direction $k^{(c)}$ is positive, return $q^{(c)}$, else return the projection of $q^{(c)}$ onto the plane defined by the direction $k^{(c)}$. It can be readily shown that VN-ReLU is rotation-equivariant (for a proof, see Appendix C.5).

**VN-Invariant layer** $\text{VN-Invariant}(\cdot\,; W) : \mathbb{R}^{C\times 3} \to \mathbb{R}^{C\times 3}$ is defined as:

$$\text{VN-Invariant}(V^{(n)}; W) \triangleq V^{(n)}\text{VN-MLP}(V^{(n)}; W)^{\intercal}, \tag{65}$$

where $\text{VN-MLP}(\cdot\,; W) : \mathbb{R}^{C\times 3} \to \mathbb{R}^{3\times 3}$ is a composition of VN-Linear and VN-ReLU layers, and $W$ is the set of all learnable parameters in VN-MLP. It can be easily shown that VN-Invariant is rotation-invariant (see Appendix C.5 for a proof).

**VN-Batch Norm, VN-Pool** For rotation-equivariant analogs of the standard batch norm and pooling operations, we point the reader to Deng et al. (2021).

### C.5 Invariance & equivariance of VN layers of Deng et al. (2021)

**Proposition 9.** *(Deng et al., 2021)* VN-Linear$(\cdot\;;W): \mathbb{R}^{C\times 3} \to \mathbb{R}^{C'\times 3}$ *is rotation-equivariant.*

*Proof.*

$$\text{VN-Linear}(V^{(n)}R;W) \triangleq WV^{(n)}R = (WV^{(n)})R = \text{VN-Linear}(V^{(n)};W)R \tag{66}$$

$\square$

**Proposition 10.** *(Deng et al., 2021)* VN-ReLU $: \mathbb{R}^{C\times 3} \to \mathbb{R}^{C\times 3}$ *is rotation-equivariant.*

*Proof.*

$$\text{VN-ReLU}(V^{(n)}R)^{(c)} \overset{(*)}{=} \begin{cases} q^{(c)}R & \text{if } \langle q^{(c)}R, k^{(c)}R\rangle \geq 0 \\ q^{(c)}R - \langle q^{(c)}R, \frac{k^{(c)}R}{||k^{(c)}R||_2}\rangle \frac{k^{(c)}R}{||k^{(c)}R||_2} & \text{o.w.} \end{cases} \tag{67}$$

$$\overset{(**)}{=} \begin{cases} q^{(c)}R & \text{if } \langle q^{(c)}, k^{(c)}\rangle \geq 0 \\ q^{(c)}R - \langle q^{(c)}, \frac{k^{(c)}}{||k^{(c)}||_2}\rangle \frac{k^{(c)}R}{||k^{(c)}||_2} & \text{o.w.} \end{cases} \tag{68}$$

$$= \left[ \begin{cases} q^{(c)} & \text{if } \langle q^{(c)}, k^{(c)}\rangle \geq 0 \\ q^{(c)} - \langle q^{(c)}, \frac{k^{(c)}}{||k^{(c)}||}\rangle \frac{k^{(c)}}{||k^{(c)}||_2} & \text{o.w.} \end{cases} \right] R \tag{69}$$

$$= \text{VN-ReLU}(V^{(n)})^{(c)}R \tag{70}$$

$$= [\text{VN-ReLU}(V^{(n)})R]^{(c)} \tag{71}$$

$$\Rightarrow \text{VN-ReLU}(V^{(n)}R) = \text{VN-ReLU}(V^{(n)})R, \tag{72}$$

where $(*)$ holds because $q$ and $k$ are rotation-equivariant w.r.t. $V^{(n)}$ and $(**)$ holds because vector inner products are rotation-invariant. $\square$

**Proposition 11.** *(Deng et al., 2021)* VN-Invariant $: \mathbb{R}^{C\times 3} \to \mathbb{R}^{C\times 3}$ *is rotation-invariant.*

*Proof.*

$$\text{VN-Invariant}(V^{(n)}R;W) = (V^{(n)}R)\text{VN-MLP}(V^{(n)}R;W)^{\mathsf{T}} \tag{73}$$

$$\overset{(*)}{=} V^{(n)}R\left[\text{VN-MLP}(V^{(n)};W)R\right]^{\mathsf{T}} \tag{74}$$

$$= V^{(n)}RR^{\mathsf{T}}\text{VN-MLP}(V^{(n)};W)^{\mathsf{T}} \tag{75}$$

$$= V^{(n)}\text{VN-MLP}(V^{(n)};W)^{\mathsf{T}} \tag{76}$$

$$= \text{VN-Invariant}(V^{(n)};W), \tag{77}$$

where $(*)$ holds by equivariance of VN-MLP. $\square$

