# OpenReview forum: "VN-Transformer: Rotation-Equivariant Attention for Vector Neurons"
_TMLR — Accepted by TMLR_

### Review · Reviewer_3fB3 · 2022-10-31

**Summary Of Contributions:**

This paper makes the following contributions, which are justified mathematically and backed by empirical results:
1. They propose rotation-equivariant attention mechanisms for vector neurons, which eliminates the need for heavy feature preprocessing.
2. They introduce the support of non-spatial attributes, which expand the applicability of the vector neuron framework.
3. They propose multi-scale feature aggregation, which greatly speeds up training and inference.
4. They show that a small bias (which introduces a small equivariance error) could accelerate hardware and improve numerical stability.
5. They demonstrate that the proposed VN-transformer can successfully be applied to 3D shape classification and motion forecasting, and empirical results support the use of the techniques mentioned above.

**Audience:**

Yes

**Broader Impact Concerns:**

I have no ethical concern over this submission.

**Claims And Evidence:**

Yes

**Requested Changes:**

1. Ideally, results presented in this paper should include standard error to account for the randomness of model initialization and optimization.
2. It may be worthwhile to discuss whether this framework can be generalized beyond rotation equivariance, maybe in future work.


**Strengths And Weaknesses:**

**Strengths:**

1. All the techniques introduced in this paper, including rotation-equivariant attention, dealing with non-spatial attributes, multi-scale feature aggregation, bias with bounded equivariance error, etc. are all very useful in my opinion, and directly address the drawbacks of the original vector neuron (VN) framework. I believe they are essential improvements to make the vector neuron framework attractive to users. The effects of these improvements are clearly demonstrated in the experiments section.
2. This paper communicates their motivation very well, and it is easy for readers to follow their math derivation. For both related work and methodology, the paper is divided into several parts, corresponding to their main contributions. So it has a good writing structure. The diagrams are also helpful.

**Weaknesses:**

This paper basically presents a batch of incremental techniques that can improve the vector neuron framework. So in terms of significance, this paper does not introduce ground-breaking results that can inspire lots of future research. (But I still think it presents many useful techniques.)

---

> ### Author Response · Authors · 2022-11-20
> **Response to Reviewer 3fB3**
>
> Thank you for the thoughtful review. We are glad you think the techniques are essential improvements on the original VN paper, that our contributions are justified empirically and mathematically, and that the paper is well-written.
>
> Regarding your comment:
>
> **"It may be worthwhile to discuss whether this framework can be generalized beyond rotation equivariance, maybe in future work."**
>
> We touch on this in the 2nd paragraph of the conclusion, where we explain that translation equivariance is achieved by simply mean-centering. In the revision, we have also added a point about scale invariance in the conclusion. If there are other types of equivariance you feel are worth mentioning, please let us know.

---

> > ### Comment · Reviewer_3fB3 · 2022-11-22
> > **Response to authors**
> >
> > After reading the authors' rebuttal, I still think this paper introduces useful techniques that can make the vector neuron framework more practical. Their claims in the abstract are well supported by theoretical and empirical results, even though their contributions seem to be incremental. I would recommend acceptance of this paper based on the technical correctness guideline of TMLR.

---

### Review · Reviewer_DJaz · 2022-11-03

**Summary Of Contributions:**

The paper extends the vector-neurons rotation equivariant point cloud network to support transformers. The suggested model is built on the facts that i) the dot product of equivariant representations can be used to model an invariant score in an attention layer, and ii) the weighted sum of equivariant representations is an equivariant representation.
Furthermore, some additions related to the Vector-Neurons framework are proposed such as the addition of bias to the linear layers and the incorporation of non-spatial features.
The model is tested on one invariant task (classification) and one equivariant task (motion prediction).


**Audience:**

No

**Broader Impact Concerns:**

The paper does not provide a broader impact statement. I do not have concerns.

**Claims And Evidence:**

No

**Requested Changes:**

Please address the weakness stated above.

**Strengths And Weaknesses:**

The proposed transformer architecture is simple and fits nicely with vector neurons.
The evaluation of the invariant task (modelnet40) supports the advantages of VN equivariant transformers versus vanilla VN.
The paper validates theoretically essential properties of the suggested model. The proofs seem to be solid.

The main observation of the paper regarding transformers (weighted sum of equivariant reps is equivariant)  has already been pointed out in [1] in more general settings. In a sense, the differences to [1] are the choice of the activation function (learnable in VN) and representation types (VN only allows type 1 and is thus simpler). Therefore, I think the authors should consider including such a discussion as part of the presentation of the method. Moreover, the evaluation should include a more detailed comparison to [1]. Currently, it only compares to a version of VN or compares to TFN and non-equivariant versions. Furthermore, the Modelnet benchmark also seems a bit odd, as the input data is aligned up to a 90 degrees rotations around the up direction. Why isn't the vnpointnet/vndgnn tested with bias?


The paper includes contributions that are independent of equivariant transformers: eps-approximation and non-spatial attributes. It is not clear why they are incorporated in an equivariant transformer paper. Moreover, the discussion on non-spatial features seems trivial, as they are merely seemed to be as invariant features.


[1]: Fabian B. Fuchs, Daniel E. Worrall, Volker Fischer, and Max Welling. Se(3)-transformers: 3d roto-translation
equivariant attention networks, 2020

---

> ### Author Response · Authors · 2022-11-20
> **Response to Reviewer DJaz**
>
> Thank you for the thoughtful review. We are glad you think the work fits nicely with vector neurons, that we validate theoretically essential properties of our model, and that our proofs are solid.
>
> We address your constructive comments point-by-point below:
>
> 1- **"The main observation of the paper regarding transformers (weighted sum of equivariant reps is equivariant) has already been pointed out in [1] in more general settings."**
>
> We do not mean to overstate the novelty of the attention mechanism (maybe this did not come through in the paper). You are right that [1] points out that an equivariant attention mechanism can be obtained from inner products between equivariant representations. We do not claim to be the first to notice this: in the revision, we have added a paragraph (see the end of Section 4.2) explaining that SE(3)-Transformer first used the idea of inner products between equivariant representations. In a way, we applied the same treatment to the VN framework as Fuchs et al. applied to the TFN framework. For reference, Appendix A also contains a detailed comparison between the two.
>
> Thank you for the suggestion, and please let us know if you think anything else should be added to the text.
>
> **2- "In a sense, the differences to [1] are the choice of the activation function (learnable in VN) and representation types (VN only allows type 1 and is thus simpler)  [...] Therefore, I think the authors should consider including such a discussion as part of the presentation of the method."**
>
> There is an important connection between (a) the TFN/SE(3)-Transformer work and (b) the VN framework. At their core, they are very similar approaches: they both represent each point in a point-cloud as a Cx3 matrix $V^{(n)}$ with a “channel” dimension and a “spatial”/”representation” dimension (whereas a traditional neural network would use a C-dimensional vector).
>
> The key difference between the approaches (a) and (b) is the way they choose to operate on and manipulate this Cx3 representation:
> (a) uses weight matrices which act on the spatial dimension of the data, i.e., right-multiplying $V^{(n)}$ with a $3\times3$ matrix $W$. In order to achieve equivariance, one must design $W$ such that $(V^{(n)}R)W = (V^{(n)}W)R$ – in other words, the matrix $W$ must satisfy $RW = WR$ for all rotations $R$ – this leads to the approach proposed by (a) which involves constructing weight matrices using Clebsch-Gordan coefficients, radial neural networks and spherical harmonics.
>
> In contrast, (b) proposes to ignore interactions between the spatial dimensions – instead they use weight matrices acting on the channel dimension of the input, i.e., left-multiplying $V^{(n)}$ with a $C’\times C$ matrix (which they call the “VN-Linear” layer). Equivariance holds trivially here ($W(VR) = (WV)R$ by associativity) for any arbitrary matrix $W$.
>
> Note that the TFN framework also proposes an operation identical to the VN-Linear layer, which they call “self-interaction.” This implies that the TFN framework is at least as expressive as (or more expressive than) the VN framework (differences between nonlinearities notwithstanding), and one could say that the VN framework is merely a “special case” of TFN.
>
> However, we believe the key advantage of the VN framework comes from its simplicity. It is far simpler to understand and implement than the TFN framework – to see this, compare the basic building block from the TFN framework (pointwise convolution layer):
>
> \begin{align}
> \mathcal{L}^{(l_o)}\_{acm_o}\big(\vec{r}_a, V^{(l_i)}\_{acm_i}\big) \triangleq \sum\_{m_f, m_i} C\_{(l_f, m_f) (l_i, m_i)}^{(l_o, m_o)} \sum\_{b\in S} F^{(l_f, l_i)}\_{c m_f}(\vec{r}\_{ab})V^{(l_i)}\_{bcm_i}
> \end{align}
>
> with the most basic layer from the VN framework:
>
> $$\textup{VN-Linear}(V^{(n)}) \triangleq WV^{(n)}.$$
>
> Again, simplicity is a key strength of VN, and we hope that improving this framework will make it even more appealing to practitioners.
>
> **"3- The paper includes contributions that are independent of equivariant transformers: eps-approximation and non-spatial attributes. It is not clear why they are incorporated in an equivariant transformer paper."**
>
> It is not clear what your suggestion is here, beyond deleting the sections on approximate equivariance and non-spatial attributes. Our goal was to construct a simple & practical rotation equivariant model, and the addition of \epsilon-approximate equivariance and non-spatial attributes is important in doing so. The motivation for each of these contributions is provided in the introduction – see paragraphs “Handling points augmented with non-spatial attributes” and “\epsilon-approximate equivariance” – please let us know if/how we can amend these paragraphs to clarify the motivation.

---

> > ### Author Response · Authors · 2022-11-20
> > **Response to Reviewer DJaz (continued)**
> >
> > **"4- Re: “Claims and Evidence” rating"**
> >
> > Your response to the “Claims & Evidence” question seems to be at odds with your statements that “the evaluation of the invariant task (modelnet40) supports the advantages of VN equivariant transformers versus vanilla VN. The paper validates theoretically essential properties of the suggested model. The proofs seem to be solid.”
> > Can you be specific about which of our claims are not supported by accurate, convincing, and clear evidence?
> >
> > **"5- Re: “Audience” rating"**
> >
> > From the TMLR Acceptance Criteria (https://jmlr.org/tmlr/acceptance-criteria.html): “If the authors make it clear that there is something to be learned by some researchers in their area from their work, then the criteria of interest is considered satisfied.”
> >
> > We think that this work would at least be of interest to the community of researchers interested in point-cloud models & equivariance as well as ML practitioners in autonomous driving.
> >
> >
> > **"6- Moreover, the discussion on non-spatial features seems trivial, as they are merely seemed to be as invariant features."**
> >
> > Your point is well-taken. Both you and reviewer x351 think that the discussion of non-spatial attributes in the main text is too simple & distracts from the more important pieces of the paper. We have significantly shortened Section 5 in the revision, and we have softened the language to de-emphasize the importance of this particular contribution.

---

### Review · Reviewer_x351 · 2022-11-15

**Summary Of Contributions:**

This paper proposes a novel rotation equivariant network -- VN-Transformer for semantic classification and motion forecasting on arbitrarily posed point clouds, whose effectiveness is verified on the popular datasets.

**Audience:**

Yes

**Broader Impact Concerns:**

More details of the investigated topic should be provided such as difference between rotation invariance and equivariance.

**Claims And Evidence:**

Yes

**Requested Changes:**

See strengths and weaknesses

**Strengths And Weaknesses:**

This topic of this paper is significant in practice, and combination of vector neurons and transformers can be an interesting attempt. The following concerns need to be revised before publishing:
1.	The authors have claimed a number of contributions in the paper, which addresses different problems for feature encoding on point clouds. The writing and organization of this paper can be less logical, and some technical proposal can be less relevant in a story of rotation-equivariant feature encoding such as non-spatial attributes.
2.	Statements in the introduction should be checked carefully, e.g. 1) it is suggested to rewrite the first paragraph to emphasize the challenge of rotation-invariance in the perspective of representation learning of 3D object shape; 2) how existing rotation-agnostic methods respectively can model rotation invariance and equivariance in the third paragraph and why?
3.	Topics investigated in the related works seem to lack of sufficient details, which makes this paper less convincing on comparison with recent works.
4.	Proof of rotation equivariant attention is given in appendix, which is suggested to place it in the main text to support the proposed method, as rotation-equivariant attention module based on Proposition 1 could be straightforward.
5.	In sec. 5, two fusion methods can be trivial, while partial rotation invariance and equivariance can be inspiring for other researchers.
6.	Content in Sec 6 and 7 seem to be implementation tricks of the proposed VN-Transformer rather than technical contributions, although they are probably important in the experiments.
7.	In the experiments, multiple settings of rotation-equivariant feature learning should be compared, such as so(3)/so(3), z/so(3); Table 2, results of VN-DGCNN should be provided.

---

> ### Author Response · Authors · 2022-11-20
> **Response to Reviewer x351**
>
> Thank you for the thoughtful review. We are glad you think the topic is practically significant, and that the combination of VN and Transformers is interesting.
>
> We address your constructive comments point-by-point below:
>
>
> **"1- The authors have claimed a number of contributions in the paper, which addresses different problems for feature encoding on point clouds. The writing and organization of this paper can be less logical, and some technical proposal can be less relevant in a story of rotation-equivariant feature encoding such as non-spatial attributes"**
>
> Your point is well-taken. Both you and reviewer DJaz think that the discussion of non-spatial attributes in the main text is too simple & distracts from the more important pieces of the paper. We have significantly shortened Section 5 in the revision, and we have softened the language to de-emphasize the importance of this particular contribution.
>
> **"2- Statements in the introduction should be checked carefully, e.g. 1) it is suggested to rewrite the first paragraph to emphasize the challenge of rotation-invariance in the perspective of representation learning of 3D object shape; 2) how existing rotation-agnostic methods respectively can model rotation invariance and equivariance in the third paragraph and why?"**
>
> Regarding 1), we have added a sentence at the end of the first paragraph to emphasize the difficulty of equipping neural networks with the right inductive biases to model 3D shapes – please let us know if anything else needs to be added.
>
> Regarding 2), we already discuss some current approaches for rotation-equivariant point cloud modeling in the “Related Work” section (see the paragraph on “Equivariant point-cloud models”) – if you think some of this should be moved to the introduction, please let us know.
>
> **"3- Topics investigated in the related works seem to lack of sufficient details, which makes this paper less convincing on comparison with recent works."**
>
> We explain in the paragraph “Equivariant point-cloud models” how the SE(3)-Transformer achieves equivariance by using weight matrices that commute with rotations (RW=WR). There is also a very detailed comparison between the two in Appendix A. Is there anything else you think should be added to the Related Work?
>
> **"4- Proof of rotation equivariant attention is given in appendix, which is suggested to place it in the main text to support the proposed method, as rotation-equivariant attention module based on Proposition 1 could be straightforward."**
>
> Point taken – in the revision, we have moved this to the main text.
>
> **"5- In sec. 5, two fusion methods can be trivial, while partial rotation invariance and equivariance can be inspiring for other researchers."**
>
> Please see our response to point 1- above – we have shortened Section 5 and softened the language to de-emphasize the importance of this contribution (we have moved the details to Appendix C).
>
> **"6- Content in Sec 6 and 7 seem to be implementation tricks of the proposed VN-Transformer rather than technical contributions, although they are probably important in the experiments."**
>
> Even if they are implementation tricks and are important in the experiments, why does that mean they are not technical contributions?

---

> > ### Comment · Reviewer_x351 · 2022-12-23
> > **My main concerns have been addressed.**
> >
> > Thanks for the reviewer's response and all my main concerns have been addressed. I recommend such a manuscript for publishing in the TMLR journal.

---

### Author Response · Authors · 2022-11-20
**Notes to all reviewers**

We thank you all for the thoughtful reviews.

1- We will write our author response in two stages:

- In order to start the discussion as soon as possible, we will first address your comments which do not involve running more experiments.

- Regarding your requests for more experiments, we are investigating what can be done in the span of the discussion period, and we will do our best to respond to these in a timely manner. In the event that we cannot complete these in time, we will incorporate the additional results into the final version of the paper.

2 - In the revised paper, we have written your requested changes in *blue* for your convenience.

---

> ### Comment · Action_Editors · 2022-11-22
> **Reviewers: please help check the authours' responses**
>
> Dear Reviewers,
>
> Thanks for reviewing the paper. The comments and suggestions are very helpful. There are some responses from authours. Could you please help check them? Thanks!
>
> best,
>
> AE

---

> > ### Author Response · Authors · 2022-11-26
> > **Polite reminder about author-reviewer discussion deadline (Nov 28th)**
> >
> > Hi everyone,
> >
> > Thank you again for the thoughtful reviews. Reviewers DJaz and x351, we wanted to kindly remind you that the author-reviewer discussion deadline is approaching (Nov. 28th). We were hoping to engage in some discussion regarding your comments, to make sure we have enough time to provide our input where it is needed. Thanks again & looking forward to discussing!
> >
> > Best,
> > Authors

---

### Decision · Action_Editors · 2022-12-27

**Recommendation:** Accept with minor revision

**Comment:**

Three experts reviewed this paper, and asked for additional clarification. After rebuttal, all three reviewers agreed with the acceptance of this paper. The AE checked the reviews and rebuttal, and would recommend the acceptance of this paper. In particular,
the authours proposed many practical techniques to address the drawbacks of the original vector neuron (VN) framework, including rotation-equivariant attention, dealing with non-spatial attributes, multi-scale feature aggregation, bias with bounded equivariance error, etc. These contributions are well supported by the experimental results.

**Audience:**

Many researchers on 3D perception and transformer-related tasks will be interested to this paper. The authours also suggested that this paper potentially will be useful to the researchers for point-cloud models & equivariance as well as ML practitioners in autonomous driving.

**Claims And Evidence:**

This paper proposes a novel rotation equivariant network -- VN-Transformer for semantic classification and motion forecasting on arbitrarily posed point clouds, whose effectiveness is verified on the popular datasets. The proposed transformer architecture is simple and fits nicely with vector neurons. It has many practical techniques to address the drawbacks of the original vector neuron (VN) framework, including rotation-equivariant attention, dealing with non-spatial attributes, multi-scale feature aggregation, bias with bounded equivariance error, etc. The methods are well evaluated on the benchmarks.

The AE recommended the acceptance of this paper (with minor revision). The authours can thus have the chance of doing the proof-reading, and finally updating the manuscript, if needed.